# GCM-SIV1.5: Optimal Tradeoff between GCM-SIV1 and GCM-SIV2

**DOI:** 10.3390/e25010107

**Published:** 2023-01-04

**Authors:** Ping Zhang

**Affiliations:** School of Computer Science, Nanjing University of Posts and Telecommunications, Nanjing 210023, China; zhgp@njupt.edu.cn

**Keywords:** nonce-based authenticated encryption, GCM-SIV1, GCM-SIV2, beyond-birthday-bound security, faulty nonce setting, mirror theory

## Abstract

GCM-SIV2 is a nonce-based beyond-birthday-bound (BBB)-secure authenticated encryption (AE) mode introduced by Iwata and Minematsu at FSE 2017. However, it is built by combining two instances of GCM-SIV1 and needs eight keys, which increases the costs of hardware and software implementation. This paper aims to reduce these costs by optimizing components (such as key materials, hash calls, and block cipher calls) and proposes an optimal tradeoff between GCM-SIV1 and GCM-SIV2 called GCM-SIV1.5. Moreover, we introduce the faulty nonce setting to AE and prove the BBB security of GCM-SIV1.5 with graceful security degradation in the faulty nonce setting by mirror theory. Finally, we discuss advantages of GCM-SIV1.5.

## 1. Introduction

The Galois Counter Mode (GCM) of operation introduced by McGrew and Viega is a very famous authenticated encryption (AE) mode [1]. Due to its friendly hardware implementation, superior software performance, no patent, and provable security, it has been widely used in high-speed network application environments. For example, GCM with the Advanced Encryption Standard (AES) has been used in IETF Transport Layer Security protocol TLS 1.3. Now, GCM has been included in the recommendations of NIST, ISO/IEC, IEEE, and IETF. As GCM is widely deployed, the CAESAR competition takes it as the baseline algorithm, which further promotes the research of GCM. There exist a large number of research results related to GCM [1,2,3,4,5,6,7,8,9,10,11,12,13,14,15,16].

GCM is a nonce-based AE mode. It takes a nonce as an extra input and requires that the nonce used in the encryption oracle is distinct (nonce-respecting setting). If the nonce length is restricted to 96 bits, GCM is provably birthday-bound secure up to approximately $2^{n/2}$ adversarial queries in the nonce-respecting setting [3,5], where *n* is the block-size of the underlying block cipher.

However, the nonce-respecting assumption does not fit the actual situation. The nonce is often misused in real life, bringing serious security threats. Joux found that, if the nonce is misused, then the hash key of GCM can be leaked and the leaked hash key can be utilized to achieve a universal forgery attack [2]. To settle the nonce misuse problem of GCM at little cost, Gueron and Lindell introduced a nonce-misuse-resistant AE (NMAE or MRAE) scheme GCM-SIV at CCS 2015 [11]. GCM-SIV covers GCM components and follows the SIV approach by Rogaway and Shrimpton [17]. In fact, as the syntax and the security model of NMAE became formalized, more and more NMAE schemes were proposed, such as [11,12,13,14,15,16,17,18,19,20,21,22,23]. GCM-SIV is just the first NMAE scheme that introduces SIV into GCM. GCM-SIV is proven secure even if the nonce is repeated. In 2016, Iwata and Minematsu pointed out that there exists a trivial distinguishing attack with approximately $2^{(n-k)/2}$ adversarial queries in GCM-SIV, where *k* is the bits of keys, and then presented an improved variant of GCM-SIV, called GCM-SIV1, which is proven secure up to $2^{n/2}$ (birthday bound) adversarial queries in the nonce misuse setting [12]. Furthermore, they considered a stronger security bound, and then proposed beyond-birthday-bound (BBB)-secure GCM-SIV*r* schemes that combine r≥2 instances of GCM-SIV1. BBB indicates that cryptographic schemes can resist beyond $O(2^{n/2})$ adversarial queries. The BBB-secure AE schemes are very rich, such as CHM [24], GCM-SIVr [12], SCT [20], ZAE [21], and PFBw [25]. GCM-SIV*r* is proven BBB-secure against $O(2^{\frac{rn}{r+1}})$ adversarial queries in the nonce misuse setting. Later, an updated variant of GCM-SIV called AES-GCM-SIV was proposed by Gueron et al., and AES-GCM-SIV was eventually accepted as a recommended standardization of IETF Crypto Forum Research Group [13,15]. Iwata and Seurin also made some important contributions to the promotion of standardization. They pointed out the problems in the earlier version, corrected them, and gave some suggestions for improving the key derivation function [14]. These problems and suggestions are accepted to further improve AES-GCM-SIV [15]. Unlike GCM-SIV, AES-GCM-SIV utilizes a key derivation function to generate the hash key and the encryption key, utilizes POLYVAL instead of GHASH, and invokes the full authentication tag as an initial counter. At Eurocrypt 2018, Bose et al. further considered the multi-user security, faster key derivation, and better bounds of AES-GCM-SIV [16].

Although there exists a large amount of research literature on the nonce misuse setting, the number of nonce misuse is often described vaguely. An effective measure of nonce misuse is the maximum number of its multi-collisions. To specify the level of nonce misuse, Dutta et al. introduced a quantitative index of nonce misuse for message authentication code (MAC) algorithms called the number of faulty nonces [23]. In the faulty nonce setting, a query is called as a faulty query if the nonce in this query is the same as the nonce in the previous queries, i.e., the nonce is re-used. The symbol μ is usually used to indicate the number of faulty nonces. Therefore, the faulty nonce setting covers nonce-respecting and nonce misuse settings. For an adversary that makes, at most, μ faulty queries, (1) if μ=0, then the adversary is called a nonce-respecting adversary; (2) if μ≥1, then the adversary is called a nonce-misusing adversary. Dutta et al. presented a nonce-based MAC scheme, nEHtM, that ensures BBB security with graceful degradation in the faulty nonce setting [23]. Furthermore, they introduced an nEHtM-based AE scheme, CWC+, whose privacy is optimally secure in the nonce-respecting setting and whose authenticity is BBB-secure with graceful degradation in the faulty nonce setting. To ensure the faulty nonce misuse resistance of privacy and authenticity, Choi et al. introduced the first fully faulty nonce-misuse-resistant AE scheme SCM [22]. It utilizes a hash key and three encryption keys. From the perspective of the security, SCM ensures close-to optimal *n*-bit security in the nonce-respecting setting and supports graceful BBB security degradation (not only for privacy but also for authenticity) in the faulty nonce setting. In recent years, the research about the faulty nonce-misuse-resistant schemes mainly focuses on MACs [26,27]. This paper aims to introduce the faulty nonce setting to GCM-SIV*r*, and presents an improved AE scheme that ensures full BBB security with graceful degradation in the faulty nonce setting and utilizes as few keys as possible.

**Our Contribution.** We focus on the optimization of GCM-SIV*r* in the faulty nonce setting, and propose an optimal tradeoff between GCM-SIV1 and GCM-SIV2 called GCM-SIV1.5, which ensures full BBB security with graceful degradation in the faulty nonce setting. Specifically, our contribution includes:From the point of view of the design, we introduce a BBB-secure sum of permutation (SoP) construction to encryption and authentication parts of GCM-SIV1.5, which makes GCM-SIV1.5 BBB secure. GCM-SIV1.5 follows “MAC-then-Encrypt” (MtE). The authentication part of GCM-SIV1.5 utilizes the construction $F_{B2}^{SoP}$ proposed by Chen et al. [27] to ensure BBB security, and the encryption part of GCM-SIV1.5 is generated by SoP-based counter mode with an initial vector and a nonce $CTR^{SoP}$ to provide BBB security. Moreover, to minimize costs of key management and implementation on software and hardware, and to maximize the running speed, GCM-SIV1.5 just utilizes two block cipher keys and a hash key, invokes a hash function and twice plaintext blocks, and generates an authentication tag. More importantly, all encryption operations involving the nonce can be carried out offline, which saves half of the online computing resources.From the point of view of the security, we prove that GCM-SIV1.5 enjoys BBB security with graceful degradation in the nonce faulty setting by using mirror theory, alternating events lemma, and the H-coefficient technique. Assuming that the underlying block cipher is a secure pseudorandom permutation (PRP) and the hash function is XOR-universal, then GCM-SIV1.5 is proven secure up to approximately 3n/4-bit query complexity and approximately *n*-bit forgery attempts for μ-nonce faulty adversaries with $\mu \leq 2^{n/4}$. In the real world, if the underlying block cipher is instantiated with AES-128, then GCM-SIV1.5 achieves, at most, approximately 96-bit security for μ-nonce faulty adversaries with $\mu \leq 2^{32}$.

In order to better demonstrate the superiority of our design, we give a fair and thorough comparison between GCM-SIV1.5 and existing typical blockcipher-based AE schemes from the following aspects: the depended assumption (PRP means pseudorandom permutation, PRF means pseudorandom function, TPRP means tweakable PRP, and ICM means ideal cipher model), the number of the encryption keys (#Encryption keys), the number of the hash keys (#Hash keys), the number of the underlying primitive (block cipher) calls (#Primitive calls), the number of the hash calls (#Hash calls), the sizes of the authentication tag and nonce, security bound under the nonce-respecting scenario (NR security), security bound under the nonce misuse scenario (NM security), and graceful degradation. The details are shown in Table 1. Compared with GCM-SIV, GCM-SIV1, GCM-SIV2, and GCM-SIV*r*, GCM-SIV1.5 utilizes fewer keys, fewer blockcipher and hash calls, and shorter sizes, provides a better security bound, and supports graceful security degradation. Therefore, GCM-SIV1.5 reduces the costs of key management and communication throughput, increases the running speed, and ensures a graceful security. Compared with CWC+, GCM-SIV1.5 provides a better security bound and supports fully faulty nonce misuse resistance and graceful security degradation for both privacy and authenticity. Compared with SCM, GCM-SIV1.5 saves an encryption key, supports offline operations involving the nonce’s encryption, and saves half of the online computing resources. In a word, our design has an excellent comprehensive performance.

The rest of this paper is organized as follows. Section 2 presents some preliminaries. Section 3 introduces mirror theory and its graph description. Section 4 shows the decomposition of nAE security. Section 5 described GCM-SIV*r*. Section 6 proposes our construction, GCM-SIV1.5. Section 7 derives the security proof. Section 8 concludes this paper.

## 2. Preliminaries

**Notations.** Some notations are described in Table 2.

**Nonce-Based Authenticated Encryption (nAE).** A nonce-based authenticated encryption (nAE) with associated data scheme Π=(K,E,D) consists of an encryption algorithm E and a decryption algorithm D, where K is a non-empty set of keys. Let K∈K. The encryption algorithm E takes a key *K*, a nonce *N*, associated data *A*, and a message *M* as the input and outputs a ciphertext and an authentication tag $\left(C,T\right)=E_{K}\left(N,A,M\right)$. The decryption algorithm D takes a key *K*, a nonce *N*, associated data *A*, a ciphertext *C*, and an authentication tag *T* as the input and outputs a message or a reject symbol $M/⊥=D_{K}\left(N,A,C,T\right)$. Here, $D_{K}\left(N,A,E_{K}\left(N,A,M\right)\right)=M$.

An nAE adversary A has access to encryption and decryption oracles $\left(E_{K},D_{K}\right)$ or random and reject oracles ($,⊥), whose goal is to distinguish them. The random oracle $ takes (N,A,M) as the input and always outputs random strings $\left(C,T\right)↞{\left\{0,1\right\}}^{\left|M|+|T\right|}$. The reject oracle ⊥ takes (N,A,C,T) as the input and always outputs a reject symbol ⊥. The nAE advantage of A against Π is defined as
$$\begin{array}{cc} Adv_{\Pi }^{nAE}\left(A\right) & =|Pr\left[K↞K:A^{E_{K},D_{K}}=1\right]-Pr\left[A^{\$,⊥}=1\right]|. \end{array}$$

We assume that A makes *q* encryption queries $\left(N^{1},A^{1},M^{1}\right),\cdots ,\left(N^{q},A^{q},M^{q}\right)$ to $E_{K}$ and returns $\left(C^{1},T^{1}\right),\cdots ,\left(C^{q},T^{q}\right)$, and then makes $q_{v}$ forgery attempts $(N^{\prime 1},A^{\prime 1},C^{\prime 1},T^{\prime 1}),\cdots$, $\left(N^{\prime q_{v}},A^{\prime q_{v}},C^{\prime q_{v}},T^{\prime q_{v}}\right)$ to $D_{K}$. For a nonce-based AE scheme, we call an AE query a faulty query if A has already queried its oracle with the same nonce, and assume that A can be allowed to make, at most, μ faulty queries. Then, μ=0 ($N^{1},\cdots ,N^{q}$ are distinct) corresponds to the nonce-respecting setting and μ≥1 (there exists at least one collision in $N^{1},\cdots ,N^{q}$) corresponds to the nonce misuse setting.

**Nonce-Based Encryption (nE).** A nonce-based encryption (nE) scheme $E=(K_{E},E-E,E-D)$ consists of an encryption algorithm E−E and a decryption algorithm E−D. The encryption algorithm E−E takes a key $K_{E}$, a nonce *N*, associated data *A*, and a message *M* as the input and outputs a ciphertext $C=E-E_{K_{E}}\left(N,A,M\right)$. The decryption algorithm E−D takes a key $K_{E}$, a nonce *N*, associated data *A*, and a ciphertext *C* as the input and outputs a message $M=E-D_{K_{E}}\left(N,A,C\right)$. Here, $E-D_{K_{E}}\left(N,A,E-E_{K_{E}}\left(N,A,M\right)\right)=M$.

An nE adversary A has access to encryption oracle $E-E_{K_{E}}$ or a random oracle $, whose goal is to distinguish them. The random oracle $ takes (N,A,M) as the input and always outputs random strings $C↞{\left\{0,1\right\}}^{\left|C\right|}$. We define the nE-advantage of A as
$$\begin{array}{c} Adv_{E}^{nE}\left(A\right)=\left|Pr\left[K_{E}↞K_{E}:A^{E-E_{K_{E}}}=1\right]-Pr\left[A^{\$}=1\right]\right|. \end{array}$$

**Pseudo-Random Function (PRF).** Let $F:K_{F}\times {\left\{0,1\right\}}^{m}\to {\left\{0,1\right\}}^{n}$ be a keyed function, where $K_{F}$ is a non-empty set of keys. It takes $K\in K_{F}$ and $X\in {\left\{0,1\right\}}^{m}$ as the input, and returns $Y=F_{K}\left(X\right)\in {\left\{0,1\right\}}^{n}$. Let R↞Func(m,n).

A PRF adversary A has access to encryption oracle $F_{K}$ or a random oracle *R*, whose goal is to distinguish them. The PRF advantage of an adversary A is defined as
$$\begin{array}{c} Adv_{F}^{prf}\left(A\right)=\left|Pr\left[K↞K_{F}:A^{F_{K}}=1\right]-Pr\left[A^{R}=1\right]\right|. \end{array}$$

**Pseudo-Random Permutation (PRP).** Let $E:K_{E}\times {\left\{0,1\right\}}^{n}\to {\left\{0,1\right\}}^{n}$ be a block cipher, where $K_{E}$ is a non-empty set of keys. It takes a key $K\in K_{E}$ and a plaintext block $M\in {\left\{0,1\right\}}^{n}$ as the input, and returns a ciphertext block $C=E_{K}\left(M\right)$. For each key $K\in K_{E}$, the function $E_{K}:{\left\{0,1\right\}}^{n}\to {\left\{0,1\right\}}^{n}$ is a permutation, i.e., $E_{K}\in Perm\left(n\right)$. Let P↞Perm(n).

A PRP adversary A has access to encryption oracle $E_{K}$ or a random permutation oracle *P*, whose goal is to distinguish them. The PRP advantage of an adversary A is defined as
$$\begin{array}{c} Adv_{E}^{prp}\left(A\right)=\left|Pr\left[K↞K_{E}:A^{E_{K}}=1\right]-Pr\left[A^{P}=1\right]\right|. \end{array}$$

**AXU Hash Functions [22,26,27,30].** Let $H:K_{H}\times {\left\{0,1\right\}}^{*}\to {\left\{0,1\right\}}^{n}$ be a hash function, where $K_{H}$ is a non-empty hash key space. Let *L* be a hash key randomly drawn from $K_{H}$. If, for any distinct $x,x^{\prime }\in {\left\{0,1\right\}}^{*}$ and $y\in {\left\{0,1\right\}}^{n}$, it holds that
$$\begin{array}{c} Pr[H_{L}\left(x\right)⊕H_{L}\left(x^{\prime }\right)=y]\leq \epsilon , \end{array}$$
then *H* is called ϵ almost XOR universal (ϵ-AXU). If $\epsilon =2^{-n}$, *H* is called an XOR universal (XU) hash function.

**Alternating Events Lemma [26,27,30].** For bounding the probability of an alternating event, such as
$$\begin{array}{c} H_{L}\left(x_{i}\right)=H_{L}\left(x_{j}\right)\wedge H_{L^{\prime }}\left(x_{j}\right)=H_{L^{\prime }}\left(x_{k}\right)\wedge H_{L}\left(x_{k}\right)=H_{L}\left(x_{l}\right), \end{array}$$
the alternating events lemma is a vital technique in the security proofs.

**Lemma** **1**(Alternating Events Lemma [26,27,30]). *Let $q_{i},q_{j},q_{k},q_{l},q$ such that $q_{i},q_{j},q_{k},q_{l}\leq q$. Let $X^{q}=\left(X_{1},\cdots ,X_{q}\right)$ be a q-tuple of random variables, and let $X^{q_{i}},X^{q_{j}},X^{q_{k}},X^{q_{l}}\subseteq X^{q}$. For distinct $i\in \left[q_{i}\right],j\in \left[q_{j}\right]$, let $E_{i,j}$ be events associated with $X_{i}\in X^{q_{i}}$ and $X_{j}\in X^{q_{j}}$, possibly dependent, which all hold with a probability of, at most, ϵ. For distinct $i\in \left[q_{i}\right],j\in \left[q_{j}\right],$$k\in \left[q_{k}\right],l\in \left[q_{l}\right]$, let $F_{i,j,k,l}$ be events associated with $X_{i}\in X^{q_{i}},X_{j}\in X^{q_{j}},X_{k}\in X^{q_{k}}$ and $X_{l}\in X^{q_{l}}$, which all hold with a probability of, at most, $\epsilon^{\prime }$. Moreover, the collection of events ${\left(F_{i,j,k,l}\right)}_{i,j,k,l}$ is independent with the collection of event ${\left(E_{i,j}\right)}_{i,j}$. Then, there exist $i\in \left[q_{i}\right],j\in \left[q_{j}\right],k\in \left[q_{k}\right],l\in \left[q_{l}\right]$ such that*
$$\begin{array}{c} Pr\left[E_{i,j}\wedge E_{k,l}\wedge F_{i,j,k,l}\right]\leq \sqrt{q_{i}q_{j}q_{k}q_{l}}\epsilon \sqrt{\epsilon^{\prime }}. \end{array}$$

**H-coefficient Technique [31].** Patarin’s H-coefficient technique is one of the very useful approaches to upper bound the distinguishing advantage of a cryptographic scheme. Given a real system *X* and an ideal system *Y*, let A be a deterministic adversary whose goal is distinguish *X* from *Y*. A interacts with *X* and *Y* and a series of query–response pairs are recorded as a transcript τ. Let T be the set of all possible transcripts. Let $X_{re}$ be the random variable interacting with the real system *X* and $Y_{id}$ be the random variable interacting with the ideal system *Y*. Then, the H-coefficient lemma is presented as follows.

**Lemma** **2**(H-coefficient Lemma). *Let $T=T_{good}\cup T_{bad}$ and ε,δ∈[0,1]. If $Pr[Y_{id}\in T_{bad}]\leq \varepsilon$ and for all $\tau \in T_{good}$, $Pr\left[X_{re}=\tau \right]/Pr\left[Y_{id}=\tau \right]\geq 1-\delta$, then*
$$\begin{array}{c} |Pr\left[A^{X}=1\right]-Pr\left[A^{Y}=1\right]|\leq \varepsilon +\delta . \end{array}$$

If an adversary makes *q* queries to an oracle *O* and obtains a transcript $\tau =\{\left(x_{1},y_{1}\right),$⋯, $\left(x_{q},y_{q})\right\}$, then we say that the oracle *O* extends the transcript τ and write it as O⊢τ, i.e., if $O\left(x_{i}\right)=y_{i}$ for all i∈[q], then O⊢τ.

## 3. Mirror Theory

Patarin’s mirror theory is a vital tool for bounding the number of solutions of affine systems of multivariate equations or non-equations, which can be applied in the security proofs of BBB-secure cryptographic schemes [27,32,33,34,35]. Here, we consider an affine system of bi-variate equations.

Let $G=<V_{1},V_{2},E,W>$ be a bipartite graph satisfying the following affine system of bi-variate equations E:$$\begin{array}{c} \left\{\begin{array}{c} X_{1}⊕Y_{1}=\lambda_{1}\\ X_{2}⊕Y_{2}=\lambda_{2}\\ \cdots \cdots \cdots \cdots \cdots \\ X_{q}⊕Y_{q}=\lambda_{q} \end{array}\right. \end{array}$$
where $X_{i}\neq Y_{j}\in {\left\{0,1\right\}}^{n}$ for any *i* and *j*, and let the vertex sets $V_{1},V_{2}$, the edge set *E*, and the weighted (labeled) function *W* be
$$\begin{array}{cc} \; & V_{1}=\left\{X_{1},,\cdots ,X_{q}\right\},V_{2}=\left\{Y_{1},\cdots ,Y_{q}\right\},\\ \; & E=\{e_{i}=\left(X_{i},Y_{i}\right),i\in \left[q\right]\},\\ \; & W:E\to {\left\{0,1\right\}}^{n}\setminus \left\{0^{n}\right\},\;and\;W\left(e_{i}\right)=\lambda_{i},i\in \left[q\right]. \end{array}$$

We assume that *G* can be divided into α components with more than two vertexes and β components with just two vertexes, i.e., $G=C_{1}\cup \cdots \cup C_{\alpha }\cup D_{1}\cup \cdots \cup D_{\beta }$.

For a bipartite graph *G*, we say that *G* is good if it satisfies the following conditions:Acylic. *G* must contain no cycle.Non-zero path label (NPL). W(P)≠0 for all paths P with an even length in the graph *G*, where $W\left(P\right)=\sum_{e\in P}W\left(e\right)$.

**Lemma** **3**(Bipartite Graph Description of Mirror Theory [27,35]). *Let $G=<V_{1},V_{2},E,W>$ be a good bipartite graph induced by E, and $|V_{1}|=q^{\prime }\leq q,|V_{2}|=q^{\prime \prime }\leq q,\left|E\right|=q$. Let $q_{c}$ be the total edges of components with more than two vertexes. Then, the number of solutions to E that are chosen from ${\left\{0,1\right\}}^{n}$ is at least*
$$\begin{array}{c} \frac{{\left(2^{n}\right)}_{q^{\prime }}{\left(2^{n}\right)}_{q^{\prime \prime }}}{2^{nq}}\left(1-\delta \right), \end{array}$$*where*
$$\begin{array}{c} \delta =\frac{9q_{c}^{2}}{8\cdot 2^{n}}+\frac{9q_{c}^{2}q+12q_{c}q^{2}+8q^{2}}{8\cdot 2^{2n}}+\frac{8q^{4}}{3\cdot 2^{3n}}. \end{array}$$

## 4. Decomposition of nAE Security

Namprempre et al. explored the generic composition of nAE and revealed the decomposition of nAE (security) from IV-based or nonce-based encryption and an MAC [36]. Now, let us focus on N3 type nAE schemes.

An N3 type nAE scheme Π=(K,E,D) consists of a PRF F and an nE scheme E, where K is the key space, E is the encryption algorithm, and D is the decryption algorithm. Given $K=\left(K_{F},K_{E}\right)\overset{\$}{\leftarrow }K=K_{F}\times K_{E}$, E takes (N,A,M) as the input and returns $\left(C,T\right)=E_{K}\left(N,A,M\right)$. To be specific, first let $T=F_{K_{F}}\left(N,A,M\right)$, and then $C=E-E_{K_{E}}\left(N,T,M\right)$. D takes (N,A,C,T) as the input and returns $M/⊥=D_{K}\left(N,A,C,T\right)$. To be specific, first let $M=E-D_{K_{E}}\left(N,T,C\right)$ and $T^{\prime }=F_{K_{F}}\left(N,A,M\right)$, and then return *M* if $T=T^{\prime }$ and ⊥ otherwise.

Type N3 nAE is secure if its tag generation function is a PRF and if the nE scheme is secure [36]. We assume that an adversary A makes, at most, *q* encryption queries and $q_{v}$ forgery attempts; then, the security of Π is shown in the following lemma.

**Lemma** **4**(Decomposition of nAE Security [36]). *Let $F:K_{F}\times N\times H\times M\to T$ be a tag generation function and $E:K_{E}\times N\times T\times M\to C$ be an nE scheme, where $T={\left\{0,1\right\}}^{\tau }$. Let Π=(K,E,D) be an N3 type nAE scheme constructed by F and E. Let A be an nAE-adversary. Then, there are two adversaries, B and C, such that*
$$\begin{array}{c} Adv_{\Pi }^{nAE}\left(A\right)\leq Adv_{F}^{prf}\left(B\right)+Adv_{E}^{nE}\left(C\right)+\frac{q_{v}}{2^{\tau }}. \end{array}$$

The above lemma shows that the security proofs of nAE schemes are reduced to the security proofs of the PRF and the nE scheme.

## 5. GCM-SIVr

Let us first review the specification of GCM-SIV*r* [12], where r≥1 is an integer. It utilizes a block cipher $E:K_{E}\times {\left\{0,1\right\}}^{n}\to {\left\{0,1\right\}}^{n}$ and a hash function $H:K_{H}\times {\left\{0,1\right\}}^{*}\to {\left\{0,1\right\}}^{n}$. The encryption algorithm E of GCM-SIV*r* takes a key $K=(L_{1},\cdots ,L_{r},K_{1}^{\prime },\cdots ,K_{r^{2}}^{\prime }$, $K_{1},\cdots ,K_{r})\in {\left(K_{H}\right)}^{r}\times {\left(K_{E}\right)}^{r^{2}+r}$, a nonce *N*, associated data *A*, and a plaintext *M* as the input, and returns a ciphertext *C* and an authentication tag $T=T_{1}\left||\cdots |\right|T_{r}$, i.e., $(C,T_{1}\left||\cdots |\right|T_{r})=E_{K}\left(N,A,M\right)$. The decryption algorithm D of GCM-SIV*r* takes *K*, *N*, *A*, *C*, and *T* as the input, and returns $M/⊥=D_{K}\left(N,A,C,T\right)$. The details are shown in Algorithms 1–5. GCM-SIV1 and GCM-SIV2 are degraded versions of GCM-SIV*r* when r=1 and 2.
**Algorithm 1** The key generation algorithm: KG**Input:** a key parameter *k***Output:** a key $K=(L_{1},\cdots ,L_{r},K_{1}^{\prime },\cdots ,K_{r^{2}}^{\prime },K_{1},\cdots ,K_{r})$$\left(L_{1},\cdots ,L_{r},K_{1}^{\prime },\cdots ,K_{r^{2}}^{\prime },K_{1},\cdots ,K_{r}\right)\overset{\$}{\leftarrow }{\left(K_{H}\right)}^{r}\times {\left(K_{E}\right)}^{r^{2}+r}$**return**$K=(L_{1},\cdots ,L_{r},K_{1}^{\prime },\cdots ,K_{r^{2}}^{\prime },K_{1},\cdots ,K_{r})$

**Algorithm 2** The encryption algorithm: E
**Input:** a key *K*, a nonce *N*, associated data *A*, and a plaintext *M*
**Output:** a ciphertext *C* and a tag *T*
Partition *M* into $M_{1}∥\cdots ∥M_{m}$, $|M_{i}\left|=n,1\leq i\leq m-1,0<\right|M_{m}|\leq n$

**for**

i=1

**to**
*r*
**do**

     Vi=HLi(N,A,M)=GHASHLi(A,M)⊕N
     Ti=0n

**endfor**


**for**

i=1

**to**
*r*
**do**

   **for**
j=1
**to**
*r*
**do**
      Ti=Ti⊕EKi+r(j−1)′(Vj)
   **endfor**

**endfor**


**for**

i=1

**to**
*r*
**do**

     $S_{i}=CTR_{K_{i}}\left(T_{i},m\right)$
     $M=M⊕msb_{\left|M\right|}\left(S_{i}\right)$

**endfor**



C←M




$T=T_{1}\left||\cdots |\right|T_{r}$



**return**

(C,T)




**Algorithm 3** The decryption algorithm: D
**Input:** a key *K*, a nonce *N*, associated data *A*, a ciphertext *C*, and a tag *T*
**Output:** a plaintext *M* or ⊥
Partition *C* into $C_{1}∥C_{2}∥\cdots ∥C_{m}$, $|C_{i}\left|=n,1\leq i\leq m-1,0<\right|C_{m}|\leq n$

**for**

i=1

**to**
*r*
**do**

     $S_{i}=CTR_{K_{i}}\left(T_{i},m\right)$
     $C=C⊕msb_{\left|C\right|}\left(S_{i}\right)$

**endfor**



M←C



**for**

i=1

**to**
*r*
**do**

     Vi=HLi(N,A,M)=GHASHLi(A,M)⊕N
     Ti=0n

**endfor**


**for**

i=1

**to**
*r*
**do**

   **for**
j=1
**to**
*r*
**do**
      Ti=Ti⊕EKi+r(j−1)′(Vj)
   **endfor**

**endfor**



$T^{\prime }=T_{1}\left||\cdots |\right|T_{r}$


**if**$T^{\prime }=T$, **return***M*
**else return** ⊥ (INVALID)

**endif**



**Algorithm 4** GHASH algorithm: $GHASH_{L}\left(A,M\right)$
**Input:** a key *L*, associated data *A*, and a plaintext *M*
**Output:** a hash value *h*
$A^{+}\leftarrow A\left|\right|0^{n-|A|\;mod\;n}$, $M^{+}\leftarrow M\left|\right|0^{n-|M|\;mod\;n}$


$X\leftarrow A^{+}\left|\right|M^{+}{\left||[|A|\right]}_{n/2}{\left||[|M|\right]}_{n/2}$


$X_{1}∥\cdots ∥X_{x}\leftarrow X$, $|X_{i}|=n,1\leq i\leq x$


h←0



**for**

i=1

**to**
*x*
**do**

      $h\leftarrow (h⊕X_{i})\cdot L$

**endfor**


**return**
*h*



**Algorithm 5** CTR algorithm: $CTR_{K}\left(T,m\right)$
**Input:** a key *K*, an initial vector *T*, and the number of plaintext blocks *m*
**Output:** a key stream *S*


$S_{1}=E_{K}\left(T\right)$



**for**

i=2

**to**
*m*
**do**

      $S_{i}\leftarrow E_{K}\left(T+i-1\right)$

**endfor**


**return**

S=S1||⋯||Sm




## 6. GCM-SIV1.5

### 6.1. Specific Description of GCM-SIV1.5

Both GCM-SIV1 and GCM-SIV2 are nonce-based authenticated encryption with associated data modes by combining a PRF and an ivE scheme. GCM-SIV1 enjoys birthday-bound security up to almost $2^{n/2}$ adversarial queries by using an *n*-bit authentication tag. GCM-SIV2 utilizes two instances of GCM-SIV1 to achieve beyond-birthday-bound (BBB) security by increasing the number of keys, authentication tags, and block ciphers. However, these methods greatly affect the implementation cost and operation efficiency of cryptographic algorithms. In real life, cryptographic algorithms that provide BBB security, as low as possible hardware and software implementation costs, and high enough operational efficiencies are much more desirable.

Given an ϵ-AXU-hash function $H:K_{H}\times N\times H\times M\to {\left\{0,1\right\}}^{n}$ and a block cipher $E:K_{E}\times {\left\{0,1\right\}}^{n}\to {\left\{0,1\right\}}^{n}$, where $K_{H}$ and $K_{E}$ are two non-empty sets of keys, and *n* is the block-size, we construct a new two-pass parallelizable nAE mode, GCM-SIV1.5. GCM-SIV1.5 is an optimal tradeoff between GCM-SIV1 and GCM-SIV2 for supporting BBB security with graceful degradation, as low as possible hardware and software implementation costs, and high enough operational efficiencies in nonce-faulty settings. We introduce a sum of permutation (SoP) construction to encryption and authentication parts of GCM-SIV1.5, which makes GCM-SIV1.5 BBB-secure. The authentication part of GCM-SIV1.5 is generated by $F_{B_{2}}^{SoP}$, which ensures BBB security. The encryption part of GCM-SIV1.5 is generated by $CTR^{SoP}$ with an initial vector and a nonce, which ensures BBB security.

The overview of GCM-SIV1.5 is illustrated in Figure 1.

GCM-SIV1.5 consists of a key generation algorithm KG, an encryption algorithm E, and a decryption algorithm D. The key generation algorithm KG takes a key parameter *k* as the input and returns a key $K=(K_{1},K_{2},L)$ (two encryption keys $K_{1},K_{2}$ and a hash key *L*) from an entropy pool of a set of keys $K=\left(K_{E},K_{E},K_{H}\right)={\left\{0,1\right\}}^{k}$. The encryption algorithm E takes a key $K=(K_{1},K_{2},L)$, a nonce *N*, associated data *A*, and a plaintext *M* as the input, invokes the tag generation algorithm $F_{B_{2}}^{SoP}$ and CTR with the SoP algorithm $CTR^{SoP}$, and outputs the corresponding ciphertext and authentication tag $\left(C,T\right)=E_{K}\left(N,A,M\right)$. The decryption algorithm D takes a key $K=(K_{1},K_{2},L)$, a nonce *N*, associated data *A*, a ciphertext *C*, and an authentication tag *T* as the input, invokes the tag generation algorithm $F_{B_{2}}^{SoP}$ and CTR with the SoP algorithm $CTR^{SoP}$, and outputs the corresponding plaintext *M* or a reject symbol ⊥, i.e., $M/⊥=D_{K}\left(N,A,C,T\right)$. The key generation, encryption, and decryption algorithms are described in Algorithms 6–8. The tag generation algorithm $F_{B_{2}}^{SoP}$ and CTR with the SoP algorithm $CTR^{SoP}$ are described in Algorithms 9 and 10.
**Algorithm 6** The key generation algorithm: KG**Input:** a key parameter *k***Output:** a key $K=(K_{1},K_{2},L)$$\left(K_{1},K_{2},L\right)\overset{\$}{\leftarrow }K=\left(K_{E},K_{E},K_{H}\right)$**return**$K=(K_{1},K_{2},L)$

**Algorithm 7** The encryption algorithm: E
**Input:** a key $K=(K_{1},K_{2},L)$, a nonce *N*, associated data *A*, and a plaintext *M*
**Output:** a ciphertext *C* and a tag *T*
Partition *M* into $M_{1}∥\cdots ∥M_{m}$, $|M_{i}\left|=n,1\leq i\leq m-1,0<\right|M_{m}|\leq n$


$T=F_{B_{2}}^{SoP}\left(K,N,A,M\right)$




$S=CTR_{K_{1},K_{2}}^{SoP}\left(N,T,m\right)$




$C=M⊕msb_{\left|M\right|}\left(S\right)$



**return**

(C,T)




**Algorithm 8** The decryption algorithm: D
**Input:** a key $K=(K_{1},K_{2},L)$, a nonce *N*, associated data *A*, a ciphertext *C*, and a tag *T*
**Output:** a plaintext *M* or ⊥
Partition *C* into $C_{1}∥C_{2}∥\cdots ∥C_{m}$, $|C_{i}\left|=n,1\leq i\leq m-1,0<\right|C_{m}|\leq n$


S=CTRK1,K2SoP(N,T,m)




$M=C⊕msb_{\left|C\right|}\left(S\right)$




$T^{\prime }=F_{B_{2}}^{SoP}\left(K,N,A,M\right)$


**if**$T^{\prime }=T$, **return***M*
**else return**⊥ (INVALID)

**endif**



**Algorithm 9** The tag generation algorithm: $F_{B_{2}}^{SoP}\left(K,N,A,M\right)$
**Input:** a key $K=(K_{1},K_{2},L)$, a nonce *N*, associated data *A*, and a plaintext *M*
**Output:** a tag *T*


$V=H_{L}\left(N,A,M\right)=GHASH_{L}\left(A,M\right)⊕N\left|\right|{\left[0\right]}_{\frac{n}{4}}$




$T=E_{K_{1}}\left(V\right)⊕E_{K_{2}}{(N||\left[0\right]}_{\frac{n}{4}})$



**return**
*T*



**Algorithm 10** CTR with SoP algorithm: $CTR_{K_{1},K_{2}}^{SoP}\left(N,T,m\right)$
**Input:** a key $K=(K_{1},K_{2})$, a nonce *N*, an initial vector *T*, and the number of plaintext blocks *m*
**Output:** a key stream *S*

**for**

1≤i≤m


     $S_{i}=E_{K_{1}}\left(T+i\right)⊕E_{K_{2}}{(N||\left[i\right]}_{\frac{n}{4}})$

**endfor**


**return**

S=S1||⋯||Sm




### 6.2. Security of GCM-SIV1.5

We present the information-theoretic security of GCM-SIV1.5 under the assumption that the underlying block cipher is a secure pseudorandom permutation.

GCM-SIV1.5 is an N3 type nAE scheme (and it can also be seen as an A7 type nAE scheme); therefore, it can be decomposed into a PRF F and an nE scheme E, where $F:K_{F}\times N\times H\times M\to T$, $E:K_{E}\times N\times T\times M\to C$, $K_{F}=K_{H}\times K_{E}\times K_{E}=K$, and $K_{E}=K_{E}\times K_{E}$.

F takes a key $K_{F}=\left(L,K_{1},K_{2}\right)\in K_{F}$, a nonce N∈N, associated data A∈H, and a message M∈M as the input and returns an authentication tag $T=F\left(K_{F},N,A,M\right)=F_{B_{2}}^{SoP}\left(K,N,A,M\right)$. E takes the key $K_{E}=\left(K_{1},K_{2}\right)\in K_{E}$, the nonce N∈N, the authentication tag T∈T, and the message M∈M as the input, computes a key-stream $S=CTR_{K_{E}}^{SoP}\left(N,T,m\right)$, and then encrypts *M* to return the corresponding ciphertext $C=E\left(K_{E},N,T,M\right)=M⊕msb_{\left|M\right|}\left(S\right)$.

According to Lemma 4, the nAE security of GCM-SIV1.5 can be decomposed into the PRF security of F and the nE security of E. Therefore, we have the following lemmas.

**Lemma** **5.**
*Let A be an μ-fault adversary and $H_{L}$ be ϵ-AXU. Let $\mu \leq q^{\frac{1}{3}}$. If A makes at most $q\leq 2^{3n/4}$ queries, then there exist adversaries $A_{1}$ and $A_{2}$ with the same query complexity against the block cipher E such that*

$$\begin{array}{cc} Adv_{F}^{prf}\left(A\right)\leq & Adv_{E}^{prp}\left(A_{1}\right)+Adv_{E}^{prp}\left(A_{2}\right)+\frac{\mu^{2}}{2^{n}}+\mu^{2}\epsilon +\frac{q^{2}\epsilon }{2^{n}}\\ \; & +4\mu^{2}\epsilon +\frac{3\mu q^{3/2}\epsilon }{2^{n/2}}+q^{4/3}\epsilon +\frac{18q^{4/3}}{2^{n}}+\frac{6q^{8/3}}{2^{2n}}\\ \; & +\frac{18q^{7/3}}{2^{2n}}+\frac{q^{2}}{2^{2n}}+\frac{8q^{4}}{3\cdot 2^{3n}}. \end{array}$$



**Lemma** **6.**
*Let A be an μ-fault adversary that makes at most $q\leq 2^{3n/4}$ queries and generates at most σ blocks, and let $\mu \leq q^{\frac{1}{3}}$ and m be the maximum block of the plaintext; then, there exist adversaries $A_{1}$ and $A_{2}$ with the same query complexity against the block cipher E such that*

$$\begin{array}{cc} Adv_{E}^{nE}\left(A\right)\leq & Adv_{E}^{prp}\left(A_{1}\right)+Adv_{E}^{prp}\left(A_{2}\right)+\frac{6m\mu^{2}}{2^{n}}\\ \; & +\frac{\sigma^{2}}{2^{2n}}+\frac{3\mu \sigma }{2^{n}}\sqrt{\frac{\sigma }{2^{n}}}+\frac{19\sigma^{\frac{4}{3}}}{2^{n}}+\frac{6\sigma^{\frac{8}{3}}}{2^{2n}}+\frac{18\sigma^{\frac{7}{3}}}{2^{2n}}\\ \; & +\frac{\sigma^{2}}{2^{2n}}+\frac{8\sigma^{4}}{3\cdot 2^{3n}}. \end{array}$$



The security proof of Lemma 5 is the same as that of Theorem 4 in the study by Chen et al. [27]. The security proof of Lemma 6 is shown in Section 7.

By combining Lemmas 4–6, we present the security of GCM-SIV1.5 as follows.

**Theorem** **1**.
*Let A be an μ-fault adversary and $H_{L}$ be ϵ-AXU. Let $\mu \leq q^{\frac{1}{3}}$ and m be the maximum block of the plaintext. If A makes at most $q\leq 2^{3n/4}$ queries and generates at most σ blocks, then there exist adversaries $A_{1}$ and $A_{2}$ with the same query complexity against the block cipher E such that*

$$\begin{array}{cc} Adv_{GCM-SIV1.5}^{nAE}\left(A\right)\leq & Adv_{E}^{prp}\left(A_{1}\right)+Adv_{E}^{prp}\left(A_{2}\right)+\frac{10m\mu^{2}}{2^{n}}\\ \; & +\frac{3q\mu \epsilon }{2^{n}}+\frac{q^{2}\epsilon }{2^{n}}+5\mu^{2}\epsilon +q^{\frac{4}{3}}\epsilon \\ \; & +\frac{(3\mu +2)\sigma }{2^{n}}+\frac{46\sigma^{\frac{4}{3}}}{2^{n}}+\frac{q_{v}}{2^{n}}. \end{array}$$



Theorem 1 shows that, if the underlying block cipher *E* is a secure PRP and $\epsilon =2^{-n}$, GCM-SIV1.5 offers BBB nAE security up to approximately $\frac{3n}{4}$-bit query complexity and approximately *n*-bit forgery attempts for μ-nonce faulty adversaries with $\mu \leq 2^{\frac{n}{4}}$.

## 7. Proofs of Lemma 6

The proof is similar to that of Theorem 4 in Chen et al. [27]. Let $K_{1},K_{2}↞K_{E}$. The adversary A makes *q* encryption queries $\left(N^{1},T^{1},m^{1}\right),\cdots ,\left(N^{q},T^{q},m^{q}\right)$ to the real world E or the ideal world *R* (*R* is an ideal version of E and always random strings) and returns $S^{1},S^{2},\cdots ,S^{q}$, and then encrypts plaintexts $M^{1},\cdots ,M^{q}$ to obtain ciphertexts $C^{1}=M^{1}⊕msb_{|M^{1}|}\left(S^{1}\right),\cdots ,$$C^{q}=M^{q}⊕msb_{|M^{q}|}\left(S^{q}\right)$. First, we replace $E_{K_{1}}$ and $E_{K_{2}}$ with two independent random permutations $P_{1}$ and $P_{2}$, and the replacements cost us $Adv_{E}^{prp}\left(A_{1}\right)+Adv_{E}^{prp}\left(A_{2}\right)$, where $A_{1}$ and $A_{2}$ are PRP adversaries against the underlying block cipher. Then, we consider $Adv_{E[P_{1},P_{2}]}^{nE}\left(A\right)$. Let $\tau =\{\left(N^{1},T^{1},m^{1},S^{1}\right),\cdots ,\left(N^{q},T^{q},m^{q},S^{q}\right)\}$. Let $X_{re}$ be the random variable interacting with the real world $X=E[P_{1},P_{2}]$ and $Y_{id}$ be the random variable interacting with the ideal world Y=R.

For the real world, the transcript with *q* queries corresponds to the following mirror system of bi-variate equations:$$\begin{array}{c} E^{=}\left\{\begin{array}{cc} \; & P_{1}\left(T^{1}+1\right)⊕P_{2}(N^{1}\left|\right|{\left[1\right]}_{\frac{n}{4}})=S_{1}^{1}\\ \; & P_{1}\left(T^{1}+2\right)⊕P_{2}(N^{1}\left|\right|{\left[2\right]}_{\frac{n}{4}})=S_{2}^{1}\\ \; & \cdots \cdots \cdots \cdots \cdots \\ \; & P_{1}\left(T^{1}+m^{1}\right)⊕P_{2}(N^{1}\left|\right|{\left[m^{1}\right]}_{\frac{n}{4}})=S_{m^{1}}^{1}\\ \; & \cdots \cdots \cdots \cdots \cdots \\ \; & P_{1}\left(T^{q}+1\right)⊕P_{2}(N^{q}\left|\right|{\left[1\right]}_{\frac{n}{4}})=S_{1}^{q}\\ \; & P_{1}\left(T^{q}+2\right)⊕P_{2}(N^{q}\left|\right|{\left[2\right]}_{\frac{n}{4}})=S_{2}^{q}\\ \; & \cdots \cdots \cdots \cdots \cdots \\ \; & P_{1}\left(T^{q}+m^{q}\right)⊕P_{2}(N^{q}\left|\right|{\left[m^{q}\right]}_{\frac{n}{4}})=S_{m^{q}}^{q} \end{array}\right. \end{array}$$As $P_{1},P_{2}$ are two independent random permutations, let $X_{i,j}=P_{1}\left(T^{i}+j\right)$, $Y_{i,j}=P_{2}(N^{i}{||\left[j\right]}_{\frac{n}{4}})$, and $\lambda_{i,j}=S_{j}^{i}$, where $j\in \left[m^{i}\right],i\in \left[q\right]$. Let $\sigma =\sum_{i=1}^{q}m^{i}$.

Let $V_{1}$ be the set of vertices $X_{1,1},\cdots ,X_{q,m^{q}}$, $V_{2}$ be the set of vertices $Y_{1,1},\cdots ,Y_{q,m^{q}}$, $E=\{e_{i,j}=\left(X_{i,j},Y_{i,j}\right),j\in \left[m^{i}\right],i\in \left[q\right]\}$, and $W:E\to {\left\{0,1\right\}}^{n}$. The above mirror system $\left\{X_{i,j}⊕Y_{i,j}=\lambda_{i,j},j\in \left[m^{i}\right],i\in \left[q\right]\right\}$ with a transcript τ can be described as an undirected weighted bipartite graph $G^{\tau }=<V_{1},V_{2},E,W>$. As *T* is random, there exist collisions in $X_{i,j}=P_{1}\left(T^{i}+j\right)$ for any $j\in \left[m^{i}\right],i\in \left[q\right]$. Let *m* be the maximum block of the plaintext. According to the fact that the nonce is μ-fault, $V_{2}$ is μ·m-fault.

In order to utilize the mirror theory, we first define a bad transcript.

**Definition** **1**(Bad Transcript). *A transcript τ is called bad if one of the following events occurs:*
*$G^{\tau }$ covers a circle of length 2 or a path of length 2 such that the weight of this path is zero.*–***B1****: There exist distinct i,k∈[q] such that $X_{i,j}=X_{k,l}$ and $Y_{i,j}=Y_{k,l}$, where $j\in \left[m^{i}\right],l\in \left[m^{k}\right]$, i.e., $T^{i}+j=T^{k}+l$ and $N^{i}{||\left[j\right]}_{\frac{n}{4}}=N^{k}\left|\right|{\left[l\right]}_{\frac{n}{4}}$ (it implies j=l).****B2****: There exist distinct i,k∈[q] such that $X_{i,j}=X_{k,l}$ and $\lambda_{i,j}⊕\lambda_{k,l}=0$, where $j\in \left[m^{i}\right],l\in \left[m^{k}\right]$, i.e., $T^{i}+j=T^{k}+l$ and $S_{j}^{i}⊕S_{l}^{k}=0$.****B3****: There exist distinct i,k∈[q] such that $Y_{i,j}=Y_{k,l}$ and $\lambda_{i,j}⊕\lambda_{k,l}=0$, where $j\in \left[m^{i}\right],l\in \left[m^{k}\right]$, i.e., $N^{i}{||\left[j\right]}_{\frac{n}{4}}=N^{k}\left|\right|{\left[l\right]}_{\frac{n}{4}}$ (it implies j=l) and $S_{j}^{i}⊕S_{l}^{k}=0$.**$G^{\tau }$ covers a path of length 4 starting at the Y-shore, or a path of length 4 starting at the X-shore such that the weight of this path is zero (this condition satisfies the fact that $G^{\tau }$ covers a circle of length 4 or a path of length 4 such that the weight of this path is zero).*–***B4****: There exist distinct i,k,w,y∈[q] such that $Y_{i,j}=Y_{k,l}$, $X_{k,l}=X_{w,x}$, and $Y_{w,x}=Y_{y,z}$, i.e., $N^{i}{||\left[j\right]}_{\frac{n}{4}}=N^{k}\left|\right|{\left[l\right]}_{\frac{n}{4}}$, $T^{k}+l=T^{w}+x$, and $N^{w}{||\left[x\right]}_{\frac{n}{4}}=N^{y}\left|\right|{\left[z\right]}_{\frac{n}{4}}$ (it implies j=l,x=z).****B5****: There exist distinct i,k,w,y∈[q] such that $X_{i,j}=X_{k,l}$, $Y_{k,l}=Y_{w,x}$, $X_{w,x}=X_{y,z}$, and $\lambda_{i,j}⊕\lambda_{k,l}⊕\lambda_{w,x}⊕\lambda_{y,z}=0$, i.e., $T^{i}+j=T^{k}+l$, $N^{k}{||\left[l\right]}_{\frac{n}{4}}=N^{w}\left|\right|{\left[x\right]}_{\frac{n}{4}}$, $T^{w}+x=T^{y}+z$, and $\lambda_{i,j}⊕\lambda_{k,l}⊕\lambda_{w,x}⊕\lambda_{y,z}=0$ (it implies l=x).**The number of edges in components with a size of more than 2 is $q_{c}\geq {\tilde{q}}_{c}$. Each vertex in the components is associated with two edges in the average case. Let us assume that it may be evenly amortized to the two vertex sets of the bipartite graph.*–***B6****: $|\left\{\left(i,k\right)|i\neq k,j\in \left[m^{i}\right],l\in \left[m^{k}\right],X_{i,j}=X_{k,l}\right\}|\geq {\tilde{q}}_{c}/4$, i.e, |{(i,k)|i≠k,$j\in \left[m^{i}\right],l\in \left[m^{k}\right],T^{i}+j=T^{k}+l\}|\geq {\tilde{q}}_{c}/4$.****B7****: $|\{\left(i,k\right)|i\neq k,j\in \left[m^{i}\right],l\in \left[m^{k}\right],Y_{i,j}=Y_{k,l}|\geq {\tilde{q}}_{c}/4$, i.e, |{(i,k)|i≠k,$N^{i}=N^{k}\}|\geq {\tilde{q}}_{c}/4$.*

*Let $\Gamma_{bad}$ be bad transcripts, *Γ* be all attainable transcripts, and $\Gamma_{good}=\Gamma \setminus \Gamma_{bad}$.*


Next, we upper bound the probability of bad transcripts in the ideal world $Pr[Y_{id}\in \Gamma_{bad}]$.

For **B1**, the probability that $T^{i}+j=T^{k}+l$ occurs for any fixed i,j,k,l is $2^{-n}$, and the number of pairs (i,k) such that $N^{i}{||\left[j\right]}_{\frac{n}{4}}=N^{k}\left|\right|{\left[l\right]}_{\frac{n}{4}}$ is at most $\mu^{2}$, where $j\in \left[m^{i}\right],l\in \left[m^{k}\right]$; then, we have
$$\begin{array}{cc} Pr[B1] & =Pr[X_{i,j}=X_{k,l},Y_{i,j}=Y_{k,l}]\\ \; & =Pr[T^{i}+j=T^{k}+l,N^{i}||{\left[j\right]}_{\frac{n}{4}}=N^{k}||{\left[l\right]}_{\frac{n}{4}}]\\ \; & \leq \frac{m\mu^{2}}{2^{n}}. \end{array}$$

For **B2**, the probability that $T^{i}+j=T^{k}+l$ occurs for any fixed i,j,k,l is $2^{-n}$, and the probability that $S_{j}^{i}⊕S_{l}^{k}=0$ occurs for any fixed i,j,k,l is $2^{-n}$; then, we have
$$\begin{array}{cc} Pr[B2] & =Pr[X_{i,j}=X_{k,l},\lambda_{i,j}⊕\lambda_{k,l}=0]\\ \; & =Pr[T^{i}+j=T^{k}+l,S_{j}^{i}⊕S_{l}^{k}=0]\\ \; & \leq \frac{\sigma^{2}}{2^{2n}}. \end{array}$$

For **B3**, the probability that $S_{j}^{i}⊕S_{l}^{k}=0$ occurs for any fixed i,j,k,l is $2^{-n}$, and the number of pairs (i,k) such that $N^{i}{||\left[j\right]}_{\frac{n}{4}}=N^{k}\left|\right|{\left[l\right]}_{\frac{n}{4}}$ is at most $\mu^{2}$, where $j\in \left[m^{i}\right],l\in \left[m^{k}\right]$; then, we have
$$\begin{array}{cc} Pr[B3] & =Pr[Y_{i,j}=Y_{k,l},\lambda_{i,j}⊕\lambda_{k,l}=0]\\ \; & =Pr[N^{i}||{\left[j\right]}_{\frac{n}{4}}=N^{k}||{\left[l\right]}_{\frac{n}{4}},S_{j}^{i}⊕S_{l}^{k}=0]\\ \; & \leq \frac{m\mu^{2}}{2^{n}}. \end{array}$$

For **B4**, the probability that $T^{k}+l=T^{w}+x$ occurs for any fixed k,l,w,x is $2^{-n}$ and the number of pairs (i,k,w,y) such that $N^{i}{||\left[j\right]}_{\frac{n}{4}}=N^{k}\left|\right|{\left[l\right]}_{\frac{n}{4}}$ and $N^{w}{||\left[x\right]}_{\frac{n}{4}}=N^{y}\left|\right|{\left[z\right]}_{\frac{n}{4}}$ for any fixed i≠k,w≠y is at most $4\mu^{2}$ (as the number of queries using any repeated nonce is at most 2μ); then, we have
$$\begin{array}{cc} Pr[B4]= & Pr[Y_{i,j}=Y_{k,l},X_{k,l}=X_{w,x},Y_{w,x}=Y_{y,z}]\\ \leq & \frac{4m\mu^{2}}{2^{n}}. \end{array}$$

For **B5**, let $F_{i,j,k,l,w,x,y,z}:\lambda_{i,j}⊕\lambda_{k,l}⊕\lambda_{w,x}⊕\lambda_{y,z}=0$, the probability that $E_{i,j,k,l}:T^{i}+j=T^{k}+l$ occurs for any fixed i,j,k,l be $2^{-n}$ (the same for $E_{w,x,y,z}:T^{w}+x=T^{y}+z$), and the probability that $F_{i,j,k,l,w,x,y,z}$ occurs for any fixed i,j,k,l,w,x,y,z be $2^{-n}$. According to alternating event lemma and σ=mq, we have
$$\begin{array}{cc} Pr[B5]= & Pr[E_{i,j,k,l},Y_{k,l}=Y_{w,x},E_{w,x,y,z},F_{i,j,k,l,w,x,y,z}]\\ \leq & \frac{3\mu \sigma }{2^{n}}\sqrt{\frac{\sigma }{2^{n}}}. \end{array}$$

For **B6**, according to Markov’s inequality, the probability of **B6** is upper bounded by
$$\begin{array}{cc} Pr[B6] & =Pr[|\left\{\left(i,k\right)|i\neq k,j\in \left[m^{i}\right],l\in \left[m^{k}\right],X_{i,j}=X_{k,l}\right\}|\geq {\tilde{q}}_{c}/4]\\ \; & \leq \frac{E[|\left\{\left(i,k\right)|i\neq k,j\in \left[m^{i}\right],l\in \left[m^{k}\right],X_{i,j}=X_{k,l}\right\}|\geq {\tilde{q}}_{c}/4]}{{\tilde{q}}_{c}/4}\\ \; & \leq \frac{\frac{\sigma^{2}}{2^{n}}}{{\tilde{q}}_{c}/4}\leq \frac{4\sigma^{2}}{{\tilde{q}}_{c}\cdot 2^{n}}. \end{array}$$

In order to obtain $\frac{3n}{4}$-bit security, we choose ${\tilde{q}}_{c}=4\sigma^{\frac{2}{3}}$. Then,
$$\begin{array}{c} Pr\left[B6\right]\leq \frac{4\sigma^{2}}{{\tilde{q}}_{c}\cdot 2^{n}}=\frac{\sigma^{\frac{4}{3}}}{2^{n}}. \end{array}$$

For **B7**, as $\mu^{2}<q^{\frac{2}{3}}\leq \sigma^{\frac{2}{3}}={\tilde{q}}_{c}/4$, the probability of **B7** being upper bounded by
$$\begin{array}{cc} Pr[B7] & =Pr[|\{\left(i,k\right)|i\neq k,j\in \left[m^{i}\right],l\in \left[m^{k}\right],Y_{i,j}=Y_{k,l}\left|\geq {\tilde{q}}_{c}/4\right]\\ \; & =Pr[\mu^{2}\geq {\tilde{q}}_{c}/4]=0. \end{array}$$

To summarize, the probability of bad transcripts is
$$\begin{array}{cc} Pr[Y_{id}\in \Gamma_{bad}] & =Pr[\overset{7}{\underset{i=1}{⋃}}\mathrm{Bi}]\\ \; & \leq \frac{6m\mu^{2}}{2^{n}}+\frac{\sigma^{2}}{2^{2n}}+\frac{3\mu \sigma }{2^{n}}\sqrt{\frac{\sigma }{2^{n}}}+\frac{\sigma^{\frac{4}{3}}}{2^{n}}. \end{array}$$

Then, we consider the ratio $\frac{Pr[X=\tau ]}{Pr[Y=\tau ]}$ between the real world *X* and the ideal world *Y* in the good transcript. In the good transcript, $G^{\tau }$ meets (1) acyclic, (2) NPL, and (3) $q_{c}\leq {\tilde{q}}_{c}=4\sigma^{\frac{2}{3}}$. Let $q^{\prime }=\left|V_{1}\right|$ and $q^{\prime \prime }=\left|V_{2}\right|$; according to the mirror theory, the number of solutions is at least $\frac{{\left(2^{n}\right)}_{q^{\prime }}{\left(2^{n}\right)}_{q^{\prime \prime }}}{2^{n\sigma }}\left(1-\delta \right)$, where
$$\begin{array}{cc} \delta = & \frac{9{\tilde{q}}_{c}^{2}}{8\cdot 2^{n}}+\frac{9{\tilde{q}}_{c}^{2}\sigma +12{\tilde{q}}_{c}\sigma^{2}+8\sigma^{2}}{8\cdot 2^{2n}}+\frac{8\sigma^{4}}{3\cdot 2^{3n}}\\ \; & =\frac{18\sigma^{\frac{4}{3}}}{2^{n}}+\frac{18\sigma^{\frac{7}{3}}+6\sigma^{\frac{8}{3}}+\sigma^{2}}{2^{2n}}+\frac{8\sigma^{4}}{3\cdot 2^{3n}}. \end{array}$$

In the real world *X*, we have
$$\begin{array}{cc} Pr[X=\tau ] & =Pr[P_{1},P_{2}\in Perm\left(n\right):E\left[P_{1},P_{2}\right]⊢\tau ]\\ \; & =\frac{|P_{1},P_{2}\in Perm\left(n\right):E\left[P_{1},P_{2}\right]⊢\tau |}{{\left|Perm\left(n\right)\right|}^{2}}\\ \; & \geq \frac{\frac{{\left(2^{n}\right)}_{q^{\prime }}{\left(2^{n}\right)}_{q^{\prime \prime }}}{2^{n\sigma }}\left(1-\delta \right)\left(2^{n}-q^{\prime }\right)!\left(2^{n}-q^{\prime \prime }\right)!}{{\left(2^{n}!\right)}^{2}}\\ \; & =\frac{1}{2^{n\sigma }}\left(1-\delta \right). \end{array}$$

In the ideal world *Y*, we have
$$\begin{array}{c} Pr\left[Y=\tau \right]=Pr\left[R\in Func\left(2n,*\right):R⊢\tau \right]=\frac{1}{2^{n\sigma }}. \end{array}$$

Therefore, the ratio between Pr[X=τ] and Pr[Y=τ] is
$$\begin{array}{c} \frac{Pr[X=\tau ]}{Pr[Y=\tau ]}\geq 1-\delta . \end{array}$$

According to the H-coefficient technique, we have
$$\begin{array}{cc} Adv_{E}^{nE}\left(A\right)\leq & Adv_{E}^{prp}\left(A_{1}\right)+Adv_{E}^{prp}\left(A_{2}\right)+\frac{6m\mu^{2}}{2^{n}}\\ \; & +\frac{\sigma^{2}}{2^{2n}}+\frac{3\mu \sigma }{2^{n}}\sqrt{\frac{\sigma }{2^{n}}}+\frac{19\sigma^{\frac{4}{3}}}{2^{n}}+\frac{6\sigma^{\frac{8}{3}}}{2^{2n}}+\frac{18\sigma^{\frac{7}{3}}}{2^{2n}}\\ \; & +\frac{\sigma^{2}}{2^{2n}}+\frac{8\sigma^{4}}{3\cdot 2^{3n}}. \end{array}$$

So far, we have completed the proof of Lemma 6.

## 8. Discussions and Conclusions

GCM-SIV1.5 is one of the favored generic nAE constructions described in [36], which combines a PRF F and an nE or ivE scheme E. Here, the PRF F is a BBB-secure $F_{B2}^{SoP}$ scheme and the nE scheme E is a BBB-secure $CTR^{SoP}$ scheme.

GCM-SIV1.5 offers an optimal tradeoff to GCM-SIV1 and GCM-SIV2 for supporting BBB security, as low as possible implementation costs, and high enough operational efficiencies. From the perspective of the security strength, if the underlying block cipher *E* is a secure PRP and $\epsilon =2^{-n}$, GCM-SIV1.5 offers approximately 3n/4-bit nAE security for μ-fault nonce-misusing adversaries and supports graceful security degradation, which is better than those of GCM-SIV1 and GCM-SIV2. From the perspective of implementation costs, compared with GCM-SIV2 and GCM-SIV*r*, GCM-SIV1.5 utilizes fewer keys (just two block cipher keys and a hash key) and lower storage and communication costs or throughput (just *n*-bit authentication tag). From the perspective of operational efficiencies, GCM-SIV1.5 utilizes just a hash function call and two plaintext blocks calls. More importantly, all encryption operations involving the nonce can be carried out offline, which saves half of the online computing resources. To sum up, our design achieves the optimal tradeoff to GCM-SIV and GCM-SIV*r* from the security strength, implementation costs, and software performance aspects.

In order to further demonstrate the superiority of our design, Table 1 shows a fair and thorough comparison between GCM-SIV1.5 and other similar schemes. Compared with CWC+, GCM-SIV1.5 provides a better security bound and supports fully faulty nonce misuse resistance, but the number of the encryption keys and the number of the block cipher calls are slightly inferior. Compared with SCM, GCM-SIV1.5 saves an encryption key, supports offline operations involving the nonce’s encryption, and saves half of the online computing resources, but other aspects, such as the number of block cipher calls, nonce size, and security bound, are slightly inferior. Besides that, SCM utilizes the finite field multiplication operations in the encryption part, although these multiplication operations can be quickly calculated using the double point technique. However, our design just utilizes some XOR and finite field addition operations.

GCM-SIV1.5 utilizes three keys. A natural future direction is to reduce the number of keys and to obtain a single-key BBB-secure variant. Besides that, GCM-SIV1.5 utilizes two plaintext blocks calls. Another future direction is to decrease the invocations of block ciphers and to improve the operational efficiencies. Our security is based on the condition that $\mu \leq 2^{n/4}$. We leave considering the case of $\mu >2^{n/4}$ as an open problem.

## Figures and Tables

**Figure 1 entropy-25-00107-f001:**
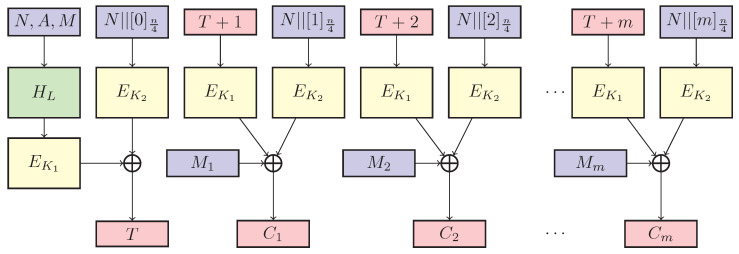
GCM-SIV1.5: An optimal tradeoff between GCM-SIV1 and GCM-SIV2.

**Table 1 entropy-25-00107-t001:** Comparison between GCM-SIV1.5 and existing typical nonce-based AE schemes, where PRP means pseudorandom permutation, PRF means pseudorandom function, TPRP means tweakable PRP, ICM means ideal cipher model, # means counting, *m* is blocks of the plaintext, *a* is blocks of associated data, and *n* is the block-size of the underlying primitive.

	Assumption	#Encryption Keys	#Hash Keys	#Primitive Calls	#Hash Calls
GCM [5]	PRP	1	1	m+1	1
ELmD [19]	PRP	1	0	a+2m+2	0
OCB3 [28]	PRP	1	1 ${}^{1}$	a+m+2	1 ${}^{2}$
ΘCB3 [28]	TPRP	1	1 ${}^{1}$	a+m+1	1 ${}^{2}$
mGCM [29]	PRP	1	1	m+1	1
GCM-SIV [11]	PRF	2	1	m+1	1
AES-GCM-SIV [15]	ICM	1 ${}^{3}$	1 ${}^{4}$	m+1	1
GCM-SIV1 [12]	PRP	2	1	m+1	1
GCM-SIV2 [12]	PRP	6	2	2m+4	2
GCM-SIV*r* [12]	PRP	$r^{2}+r$	*r*	$rm+r^{2}$	*r*
CWC_+_ [23]	PRP	1	1 ${}^{5}$	m+3	1
SCM [22]	PRP	3	1	m+5	1
GCM-SIV1.5	PRP	2	1	2m+2	1
	**Tag Size**	**Nonce Size**	**NR Security**	**NM Security**	**Graceful Degradation**
GCM [5]	≤*n*	3n/4	$O(2^{n/2})$	-	×
ELmD [19]	*n*	*n*	$O(2^{n/2})$	$O(2^{n/2})$	×
OCB3 [28]	≤*n*	≤*n*	$O(2^{n/2})$	-	×
ΘCB3 [28]	≤*n*	≤*n*	$O(2^{n})$	-	×
mGCM [29]	*n*	*n*	$O(2^{n})$	-	×
GCM-SIV [11]	*n*	*n*	$O(2^{n/2})$	$O(2^{n/2})$	×
AES-GCM-SIV [15]	*n*	3n/4	$O(2^{3n/4})$	$O\left(2^{n/2}\right)∼O\left(2^{3n/4}\right)$	✔
GCM-SIV1 [12]	*n*	*n*	$O(2^{n/2})$	$O(2^{n/2})$	×
GCM-SIV2 [12]	2n	*n*	$O(2^{2n/3})$	$O(2^{2n/3})$	×
GCM-SIV*r* [12]	rn	*n*	$O(2^{rn/r+1})$	$O(2^{rn/r+1})$	×
CWC_+_ [23]	≤*n*	3n/4	$O(2^{2n/3})$	$O\left(2^{n/2}\right)∼O\left(2^{2n/3}\right){\;}^{6}$	✔
SCM [22]	*n*	n−2	$O(2^{n})$	$O\left(2^{n/2}\right)∼O\left(2^{n}\right)$	✔
GCM-SIV1.5	*n*	3n/4	$O(2^{3n/4})$	$O\left(2^{n/2}\right)∼O\left(2^{3n/4}\right)$	✔

^1^ The hash key is the encryption key. ^2^ The hash function is achieved by invoking *a* underlying primitives. ^3^ The encryption key is generated by invoking a key derivation function. ^4^ The hash key is generated by invoking a key derivation function. ^5^ The hash key is generated by the encryption key. ^6^ This security bound is just that of authenticity. The privacy of CWC_+_ is insecure in the nonce misuse setting.

**Table 2 entropy-25-00107-t002:** Descriptions of notations.

Notations	Descriptions
⊕	the bitwise exclusive or (XOR)
+	addition modulo $2^{n}$
·	the multiplication over the finite field
||	the concatenation of strings
${\left\{0,1\right\}}^{*}$	a set of all strings (including an empty string)
${\left\{0,1\right\}}^{n}$	a set of all strings whose bit-length is *n*
Perm(n)	a set of all permutations whose workspace is *n*
Func(m,n)	a set of all functions from *m*-bit inputs to *n*-bit outputs
K↞K	the key *K* randomly sampled from the key space K
$A^{O}=1$	an event where an adversary A outputs 1 after interacting with the oracle *O*
${\left[i\right]}_{m}$	an *m*-bit binary representation of an integer *i*
[r]	a set of consecutive integers {1,2,⋯,r}
|X|	the number of elements in the set *X*
${\left(2^{n}\right)}_{q}$	$2^{n}\cdot \left(2^{n}-1\right)\cdots \left(2^{n}-q+1\right)$

## Data Availability

The data used to support the findings of the study are available within the article.

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
