# Peer review of "GCM-SIV1.5: Optimal Tradeoff between GCM-SIV1 and GCM-SIV2"

_entropy, 2023, doi:10.3390/e25010107_

Round 1
Reviewer 1 Report
The authors have presented a modified authenticated encryption (AE) scheme GCM-SIV1.5 that is based on GCM-SIV1 and GCM-SIV2 and claim that the proposed scheme is more efficient in terms of software and hardware implementation. The paper is well-written and mathematical proofs of the scheme have been presented to justify the claims. Still, there are some minor concerns that need to be answered:
1. The discussion and conclusion section must be updated to show the proposed scheme's efficiency in comparison to other similar schemes clearly.
2. A high level of plagiarism is evident may be due to very similar schemes presented but need to be handled in line with publisher guidelines.
Author Response
- Response to Comments: The authors have presented a modified authenticated encryption (AE) scheme GCM-SIV1.5 that is based on GCM-SIV1 and GCM-SIV2 and claim that the proposed scheme is more efficient in terms of software and hardware implementation. The paper is well-written and mathematical proofs of the scheme have been presented to justify the claims. Still, there are some minor concerns that need to be answered: 1. The discussion and conclusion section must be updated to show the proposed scheme's efficiency in comparison to other similar schemes clearly.
Response: Thanks very much for your recognition of our work and your good comments. We have refined Table 1 to show the comparison between our scheme and other similar schemes in the introduction section. Meanwhile, the detailed comparison results are also described in the discussion and conclusion section.
- Response to Comments: 2. A high level of plagiarism is evident may be due to very similar schemes presented but need to be handled in line with publisher guidelines.
Response: We have restated the relevant sections of this article, and have added references to reduce the repetition rate and avoid plagiarism. Thanks.

Reviewer 2 Report
This paper proposes a new nonce-misuse resistant AEAD scheme, called GCM-SIV1.5. The construction can be seen as a refinement of GCM-SIV1 and GCM-SIV2. The authors prove that in the faulty-nonce setting GCM-SIV1.5 achieves 3n/4-bit security while the number of faulty-nonce queries is at most 2^{n/4}.
This is a moderately interesting result. The authors apply the N3 composition from the seminal (Namprempre et al., Eurocrypt 2014). For the PRF part the authors use the existing SoP like construction introduced in (Chen et al., Asiacrypt 2021), and the encryption part is a similar application of SoP with tag and nonce as inputs. The results are not surprising and having a good knowledge of the state-of-the-art, I think the main security result is correct, at least in asymptotic sense.
Apart from the lack of novelty, I also observed that the paper fails to cite some of the important related works. Alas! the alternating events lemma is wrongly attributed to Chen et al., when in fact it was introduced by Jha and Nandi in (Jha and Nandi J. Crypto. 2020). Indeed, even Chen et al. attribute the result to Jha and Nandi. Similarly, there is no mention of the SCN construction (Choi et al. Asiacrypt 2021), which bears a very close resemblance to the proposed construction in both design and features, and hence demands a fair and thorough comparison.
Overall, I think the paper lacks the required technical novelty and editorial maturity to demand publication at this moment. I recommend rejection.
Author Response
- Response to Comments: This paper proposes a new nonce-misuse resistant AEAD scheme, called GCM-SIV1.5. The construction can be seen as a refinement of GCM-SIV1 and GCM-SIV2. The authors prove that in the faulty-nonce setting GCM-SIV1.5 achieves 3n/4-bit security while the number of faulty-nonce queries is at most 2^{n/4}. This is a moderately interesting result. The authors apply the N3 composition from the seminal (Namprempre et al., Eurocrypt 2014). For the PRF part the authors use the existing SoP like construction introduced in (Chen et al., Asiacrypt 2021), and the encryption part is a similar application of SoP with tag and nonce as inputs. The results are not surprising and having a good knowledge of the state-of-the-art, I think the main security result is correct, at least in asymptotic sense. Apart from the lack of novelty, I also observed that the paper fails to cite some of the important related works. Alas! the alternating events lemma is wrongly attributed to Chen et al., when in fact it was introduced by Jha and Nandi in (Jha and Nandi J. Crypto. 2020). Indeed, even Chen et al. attribute the result to Jha and Nandi…
Response: We are very sorry for the lack of important literature, our misquotation, and unclear writing. We have added some important references and corrected the wrong misquotation. The previous version of our manuscript focused on the optimization of GCM-SIVr in the faulty nonce setting. Therefore, it just considered the comparison among GCM-SIV, GCM-SIV1, GCM-SIV2, GCM-SIVr and our design GCM-SIV1.5, which is not comprehensive. From the goal point of view, our design achieves the optimal tradeoff to GCM-SIV and GCM-SIVr (r>=1) from the security strength, implementation costs, and software performance aspects. GCM-SIV1.5 supports BBB security with graceful security degradation in the faulty nonce setting, utilizes fewer keys and lower storage and communication costs or throughput, supports offline operations involving the nonce's encryption, and saves half of the online computing resources. From the design point of view, GCM-SIV1.5 is innovative. We introduce offline operations involving the nonce's encryption to implement preprocessing and support BBB-secure encryption and authentication parts.
- Response to Comments: This paper proposes a new nonce-misuse resistant AEAD scheme, called GCM-SIV1.5...This is a moderately interesting result…Similarly, there is no mention of the SCN construction (Choi et al. Asiacrypt 2021), which bears a very close resemblance to the proposed construction in both design and features, and hence demands a fair and thorough comparison. Overall, I think the paper lacks the required technical novelty and editorial maturity to demand publication at this moment. I recommend rejection.
Response: Thanks for your good comments. We have updated Table 1 and have added a fair and thorough comparison between our scheme and other similar schemes. Indeed, we just focused on the comparison among GCM-SIV, GCM-SIV1, GCM-SIV2, GCM-SIVr and our design GCM-SIV1.5 before, and omitted other similar schemes, such as SCM (Choi et al. Asiacrypt 2021), which is not comprehensive. For the innovation of our manuscript and the comparison with SCM scheme, we make the following detailed descriptions. From the design point of view, GCM-SIV1.5 is innovative. We introduce offline operations involving the nonce's encryption to implement preprocessing, which is better than other similar schemes including GCM-SIV2 and SCM. Furthermore, our design achieves the optimization of GCM-SIVr in the faulty nonce setting. Compared with SCM, GCM-SIV1.5 saves an encryption key, supports offline operations involving the nonce's encryption, and saves half of the online computing resources, but other aspects, such as the number of the block cipher calls, the nonce size, and the security bound, are slightly inferior. Besides that, SCM utilizes the finite field multiplication operations in the encryption part, although these multiplication operations can be quickly calculated using the double point technique. However, our design just utilizes some XOR and finite field addition operations. As a result, we feel our design is innovative and meaningful.

Round 2
Reviewer 2 Report
The revised manuscript has satisfactorily incorporated my previous comments. Barring block cipher keys, the proposed construction is mostly inferior to the best state-fo-the-art construction, i.e., SCM. Nevertheless, I recommend acceptance, for the sake of variety among constructions.